

**Main drivers of plant diversity patterns of rubber plantations in the**
**Greater Mekong Sub-region**
Guoyu Lan[a,c#*], Bangqian Chen[a,c#], Chuan Yang [a,c], Rui Sun [a,c], Zhixiang Wu [a,c], Xicai
Zhang[a,c]
a. Rubber Research Institute, Chinese Academy of Tropical Agricultural Sciences, Danzhou
City, Hainan Province, 571737, P. R. China;
c. Danzhou Investigation & Experiment Station of Tropical Crops, Ministry of Agriculture
and Rural Affairs, Danzhou City, Hainan Province, 571737, P. R. China
#These authors contributed equally to this work.
*Author for correspondence
Tel: +86-898-23301800
Fax: +86-898-23300315
E-mail: langyrri@163.com
Running headline: Drivers of plant diversity of rubber plantations

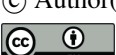



**Abstract:**
The Greater Mekong Sub-region (GMS) is one the global biodiversity hotspots. However, the
diversity has been seriously threatened due to environmental degradation and deforestation,
especially by expansion of rubber plantations. Yet, little is known about the impact of rubber
plantations on plant diversity. In this study, we analyzed plant diversity patterns of rubber
plantations in the GMS based on a ground survey of a large number of samples. We found
that diversity varied across countries due to varying agricultural intensities. Laos had the
highest diversity, then followed China, Myanmar, Cambodia. Thailand and Vietnam were the
lowest among them. Plant species richness of Laos was about 1.5 times that of Vietnam. We
uncovered latitudinal and longitudinal gradients in plant diversity across these artificial
forests of rubber plantations. These gradients could be explained by the traditional ecological
theories. Furthermore, null deviation of observed community to the randomly assembled
communities were larger than zero indicating deterministic process were more important for
structuring the community. Meanwhile, the results also showed that higher dominance of
some exotic species (such as *Chromolaena odorata* and *Mimosa pudica*) were associated
with a loss of plant diversity within rubber plantations. In conclusion, not only environmental
factors (such as elevation and latitude), but also exotic species were the main factors affecting
diversity of these artificial stands. Much more effort should be made to balance agricultural
production with conservation goals in this region, particularly to minimize the diversity loss
in Vietnam and Cambodia.
**Keywords:** Rubber plantation, Plant diversity, Exotic species, Mekong regions, Greater
Mekong Sub-regions (GMS)




## 1. Introduction

The Great Mekong Sub-region (GMS) is one of the most important biodiversity hotspots in

the world (Myers et al., 2000). It covers Yunnan province of south China, Thailand, Vietnam,

Cambodia, Laos, and Myanmar. Conservation and management of forests in this area are

difficult due to conflicting external social and economic factors. Cambodia, Laos, and

Myanmar have been recognized among the least developed countries in the world by the

United Nations. Meanwhile, the urban and rural development of Vietnam and Thailand is

unbalanced, and there are still a large number of population under poverty line. Recently, the

GMS has been identified as a major strategic source of raw, extractable materials in Asia

(Zhou and Wei, 2009).

In the GMS, logging, mining, and slash and burn agriculture contribute to deforestation

and forest degradation. Much of the land has recently been converted from forest to

agriculture (Li et al., 2007), and rubber plantations have quickly expanded throughout the

region (Ziegler et al., 2009; Li et al., 2015). A large area of natural forest has been replaced

by rubber plantations (Ahrends et al., 2015) due to a surge in the global demand for natural

rubber, driven largely by the growth of tire and automobile industries. For example, 23.5% of

Cambodia's forest cover – more than 2.2 million hectares –was destroyed between 2001 and

2015 make way for crops such as rubber (Figure S1h) and palm oil (Grogan et al., 2019).

Almost one-quarter of cleared land has been used for plantations of the non-native rubber

tree. In southwest China, nearly 10% of the total area of nature reserves had been converted

to rubber monoculture by 2010 (Chen et al., 2016). At present, GMS are globally important



rubber-planting regions (Xiao et al., 2021). Though rubber expansion caused deforestation,
cultivated rubber plantations have helped alleviate poverty in low-income regions, and rubber
cultivation is the main economic source of farmers in remote areas in some areas of the GMS,
such as Laos, Myanmar and Cambodia (Figure S1c-e).

Agricultural land-uses can exacerbate many infectious diseases in Southeast Asia (Shah et

al., 2019) and reduce biodiversity (Fitzherbert et al., 2018; Zabel et al., 2019; Singh et al.,
2019). Expansion of rubber plantations is a resurgent driver of deforestation, carbon
emissions, and biodiversity loss in this region (Xu, 2011; Warren-Thomas et al., 2018).
It is indisputable that the large-scale rubber cultivation in countries of the GMS has an
outsized impact on the ecosystem of tropical regions. There is also a large body of literature
on the effects of forest conversion from tropical forest to rubber plantations on soil microbial
composition and diversity (Schneider et al., 2015; Kerfahi et al., 2016, Lan et al., 2017a;
2017b; 2017c; Lan et al., 2020a; 2020b; 2020c). Compared to primary forests, agricultural
systems tend to have higher bacterial richness but lower fungal richness (Lan et al., 2017a;
Cai et al., 2018; Tripathi et al., 2012; Kerfahi et al., 2016). Forests that are intensively
managed for production purposes generally have lower biodiversity than natural forests
(Chaudhary et al., 2016), and this is especially true for rubber plantations (He and Martin,
2016). Plant diversity of artificial forests is greatly affected by agricultural and management
activities, such as the application of herbicides and sprout control. However, there were few
reports on the effects of forest conversion on plant diversity in this region. We do not know,
for example, if there are differences in plant diversity among countries and how exotic plants
may affect local plant diversity in rubber plantations? Two types of processes, deterministic



(Lan et al., 2011) and stochastic (Hu et al., 2012), not only affect tropical forest plant
community assembly, but also microbial assembly (Stegen et al., 2012; Zhou et al., 2014).
However, the relative influences of the two processes on plant community for rubber
plantation and drivers of plant diversity are still unclear. To address these questions, we
surveyed a large number of plots on rubber plantations in the GMS to investigate plant
diversity and associated causes. Our study provides an empirical case for understanding the
effect of rubber plantations on plant diversity in the Greater Mekong region and the
restoration and protection of biodiversity in this region.

**2.  Methods**
*2.1 Study area*
The Mekong River Basin has a total length of 4880 km and a drainage area of 795000 square
kilometers, with 326 million people living in the basin. The GMS encompasses a variety of
climate types and geographical characteristics, and is rich in water and biological resources
(Wu et al., 2020). Rubber plantations are one of the most widespread vegetation types in the
region, and are distributed throughout the south of Yunnan province, almost all states of
Thailand and Laos, the southern half of Vietnam and Myanmar, and the eastern half of
Cambodia.



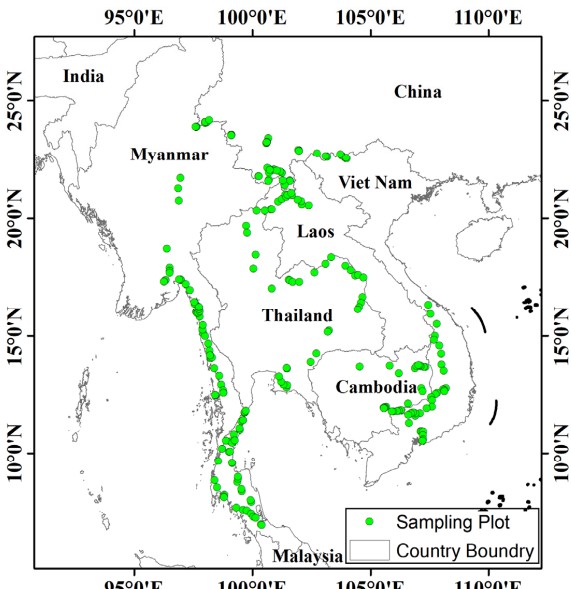

**Figure 1** Sampling plot localities within rubber plantations in GMS

### 2.2 Sampling methods

Before the field investigation, we first determined the investigation route according to the distribution of rubber plantation in this regions. Then, plots were randomly selected approximately equidistant from each other (every 10-20 km according to the actual situation) along the investigation route (Yaseen, 2013). We did not deliberately select plots according types of rubber plantation, thus these plots were independent from each other. Consequently, a total of 240 plots, each with an area of 100 m² (10 m × 10 m), were selected in the GMS, with 32 plots in Vietnam, 24 in Cambodia, 15 in Laos, 73 in Thailand, 47 in Myanmar, and 56 in China (Figure 1). We started the investigation only after the guide (local people) asked the farmer's consent. Plot measurements, such as longitude, latitude, elevation, slope degree, slope aspect, rubber tree height, and canopy density were recorded in detail (Table S1). Annual and





perennial plant species, shrubs, trees and lianas as well as theirs seedlings were recorded. We
do not investigate bryophytes, but ferns were investigated. Species information, such as species
name, height and coverage, life form (non-woody, shrub, liana or tree) (Lan et al., 2014), from
each plot in the rubber plantations were also recorded. We visually assigned a cover value to
each species in each quadrant of the plot, using an ordinal cover class scale with class limits
0.5%, 1%, 2%, 5%, 10%, 15%, 20%, and thereafter every 10% up to 100%. The cover values
for each species in the plot were then averaged across the four quadrants (Sabatini et al., 2016).
Climate data, including annual average temperature and annual average precipitation, were
obtained from WorldClim (http://worldclim.org) based on the geographic coordinates of each
sample site.
**_2.3 Data analysis_**
According to Sun's ( 2000) classification of plant uses, species were divided into medicinal
plants, edible plants, economic plants, forage plants, ornamental plants, ecological plants and
others (unknown use). Relative height (*RH*), relative dominance (*RD*, using coverage), and
relative frequency (*RF*) were calculated for each species to estimate the importance value
(*IV*). Importance value, as defined here, differs from previous studies (e.g., Curtis and
Mcintosh 1950, 1951; Greig-Smith 1983; Linares-Palomino and Alvarez 2005) because most
understory species are herbs, which make precise measure of abundance difficult. We define
the importance value as:
Importance value: $IV_j = RF_j + RH_j + RD_j$,
Relative frequency: $RF_j = 100 \times F_j / \sum_j F_j$
Relative height: $RH_j = 100 \times H_j / \sum_j H_j$,





Relative dominance: $RD_j = 100 \times D_j / \sum_j D_j$
where $F_j$ was the number of plots containing species $j$; $D_j$ was the coverage of species $j$; and
$H_j$ was the height of species $j$.

Species richness, the Shannon index were used to measure α diversity of each plot. It should

be noted that the importance values of each species were used to calculate the Shannon
diversity (i.e., replace "abundance" or "number of individuals" with "important value").
Whittaker's β diversity was used to estimate the diversity across different countries and was
calculated as follows (Whittaker, 1960)

$\beta w = S/m_a - 1$

where $S$ is the total species richness of all samples and $m_a$ is the mean species richness of
these samples. Principal coordinates analysis (PCoA) based on Bray–Curtis distance of
species IVs (importance values) was performed to compare plant species composition across
countries using R package "amplicon". Analysis of similarity (ANOSIM) was used to test for
differences in diversity indices among study sites. Linear regression was used to find whether
there were positive or negative correlations between diversity (richness, Shannon index) and
environmental variables including latitude, longitude, elevation, rainfall, temperature, slope
degree, tree age, tree height as well as canopy density.

Machine learning algorithm, Random forests, was used to rank the feature importance of

environmental factors with 999 iterations. To evaluate the influences of the neutral processes
on plant community of rubber plantation, the null deviation was measured as the difference of
the β diversity (i.e., Bray-Curtis dissimilarity) between the observed and randomly plant
communities. A null deviation of zero indicates that the communities follow the stochastic or





near-stochastic distribution, whereas a null deviation larger than zero indicates that
deterministic processes cause the communities to be more dissimilar than null expectations
(Liu et al., 2021, Zhou et al., 2014). Null deviations were calculated for plant communities
across six countries from 1000 stochastic assemblages (Lee et al., 2017).

**3 Results**
***3.1 Plant composition of rubber plantations***

A total of 949 plant species, representing 550 genera and 153 families, were recorded

across rubber plantations of the six countries (Table 1 &Table S2). There are 597 species of
medicinal plants, 163 species of edible plants, 220 species of economic plants, 64 species of
forage plants, 158 species of ornamental plants, 62 species of ecological plants, and 170 species
of unidentified uses under rubber plantation in GMS (Table S3). Our results also showed that
445 (46.89%) were herbs, with a largest number of Compositae (Table S4). Plant communities
of rubber plantation tended to be dominated by Fabaceae, Euphorbiaceae, Poaceae, Rubiaceae,
and Compositae (Table S4). The five most common species observed were *Cyrtococcum patens,*
*Chromolaena odorata, Asystasia chelonoides, Axonopus compressus,* and *M. pudica* (Table
S5). 237 plots containing exotic plant species, most of them were from tropical America. A
total of 121 (12.75%) species were identified as exotic (belonging to 45 families and 91 genera).
The five most common exotic species were *C. odorata*, *M. pudica*, *Axonopus compressus*,
*Ageratum conyzoides*, and *Borreria latifolia.*  *C. odorata* and *M. pudica* were recorded in
almost every plot (Figure 2). The exotic species richness of rubber plantations was relatively
higher in Cambodia, Vietnam, and Myanmar compared to China, Laos, and Thailand (Figure





3c).

Table 1 Composition of plants of rubber plantations in GMS

| Types | No. of | Lifeform (%) | No. of families | No. of genera | No. of species |
|---|---|---|---|---|---|
| Ferns | 76 (8.00) | Non-woody plant | 86(38.05) | 278(45.65) | 445(46.89) |
| Gymnosperms | 3 (0.32) | Liana | 32(14.16) | 62(10.18) | 101(10.64) |
| Angiosperm | 870 (91.68) | Shrub | 42(18.58) | 118(19.38) | 192(20.23) |
| | | Tree | 66(29.20) | 151(24.79) | 211(22.23) |
| Total | 949 (100) | Total | 226(100.00) | 609 | 949 |


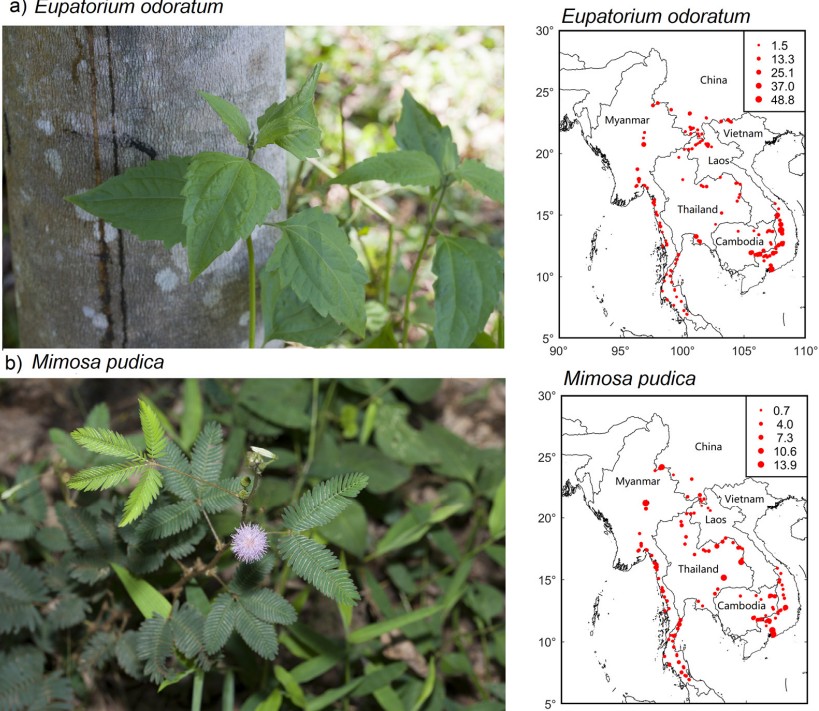


**Figure 2** Distribution maps of two common exotic species (a: *Chromolaena odorata*, b:
*Mimosa pudica*) of rubber plantation in the GMS (circle size is proportional to importance
value)
***3.2 Plant diversity of rubber plantations***





Species richness of rubber plantations in Laos was the highest among the six countries,
followed by China and Myanmar, while the richness of Thailand, Cambodia, and Vietnam
were relatively lower (Figure 3a). The same was true for Shannon diversity diversity (Figure
3b). PCoA plots showed significant differences in species composition among some countries
(Figure 4a). Beta diversity among countries showed that Cambodia and Vietnam had similar
species compositions, as did Thailand and Myanmar (Figure 4b). The beta diversity between
China and other countries was consistently high.

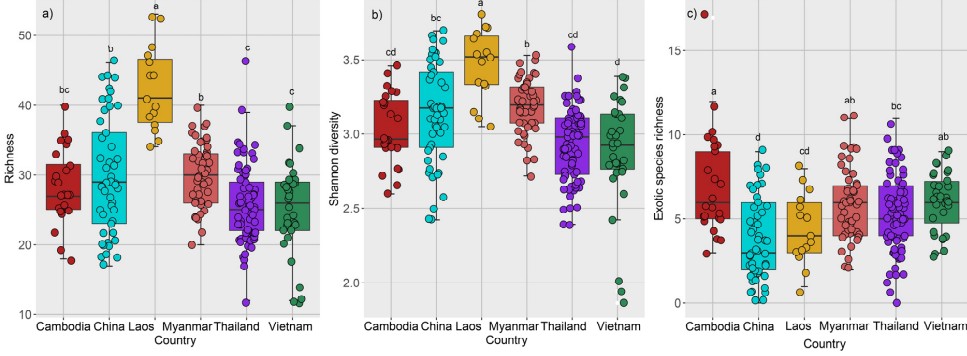


**Figure 3** Plant species diversity of rubber plantations across countries in the GMS (a: species
richness; b: Shannon diversity; c: Exotic species richness).

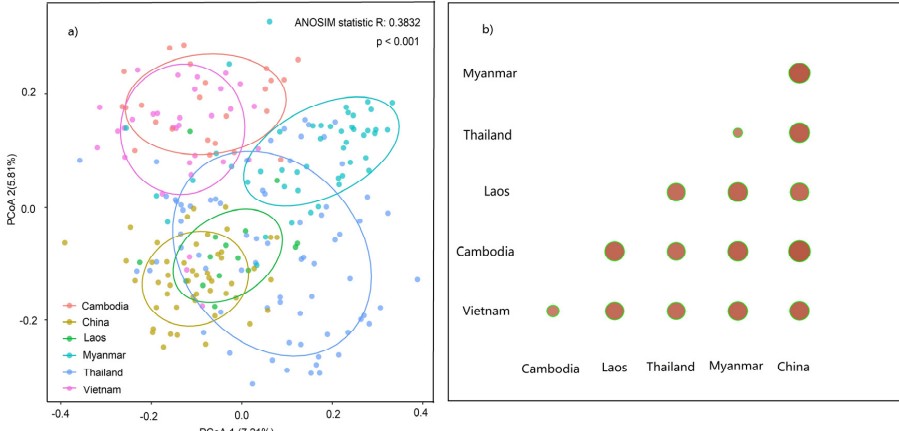






**Figure 4** Beta diversity of rubber plantations in the GMS (a: PCoA ordination plot, b:
Whittaker's beta diversity (circle size is proportional to beta diversity value))

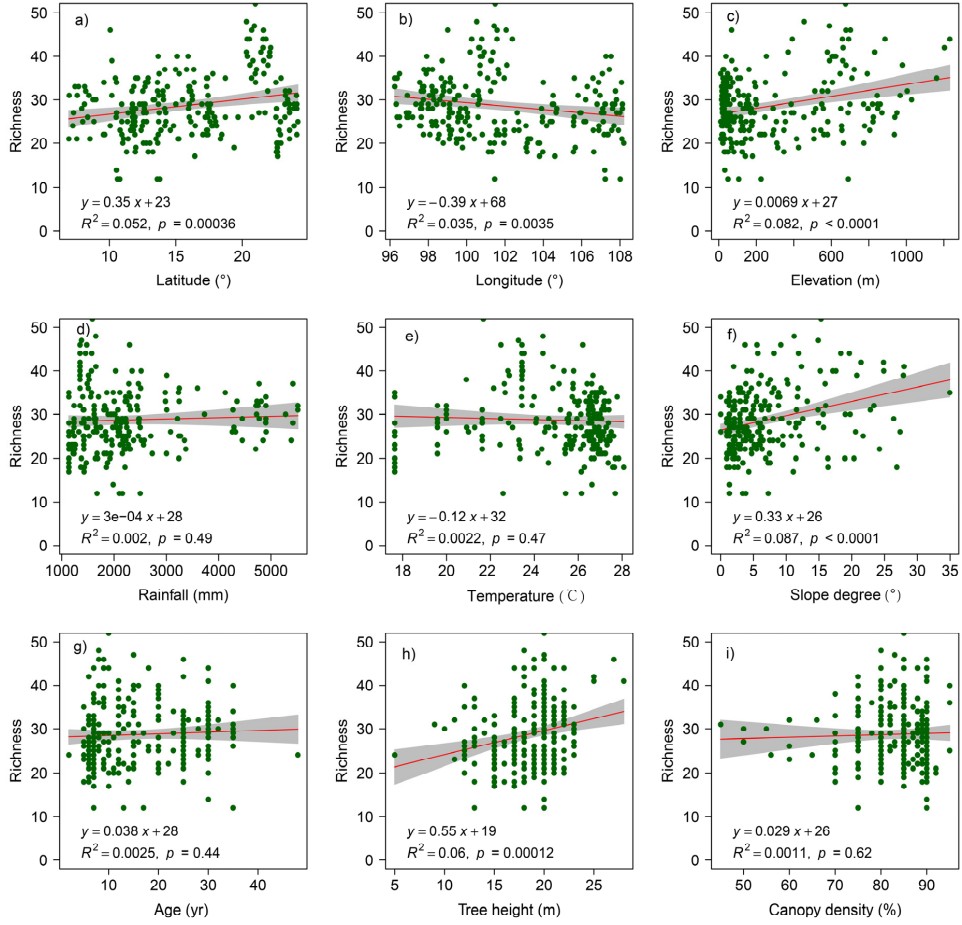


**Figure 5** Linear regressions of species richness of rubber plantation with environmental
variables (a: latitude; b: longitude; c: elevation; d: rainfall; e: temperature; f: slope degree; g:
tree age, h: tree height; i: canopy density)

Diversity was significantly correlated with latitude, longitude, and elevation (Figure 5a-c).
Linear regressions showed that diversity indices of richness, Shannon diversity, and Simpson





diversity significantly increased with latitude and elevation ($p < 0.05$), however deceased
with longitude ($p < 0.001$). Slope ($p < 0.001$) (Figure 5f) and tree height ($p < 0.001$) (Figure
5h) also were also important factors influencing diversity of rubber plantations. Rubber tree
ages, canopy density, rainfall, and temperature showed no effects on diversity ($p > 0.05$).
Random forest results showed that high mean squared errors of latitude, longitude, and
countries were the top three features affecting plant diversity of rubber plantation (Figure 6a).
The null deviations were greater than 0.5 for all these six countries (Figure 6b) indicating
deterministic process were more important than stochastic process for the assembly of this
artificial system.

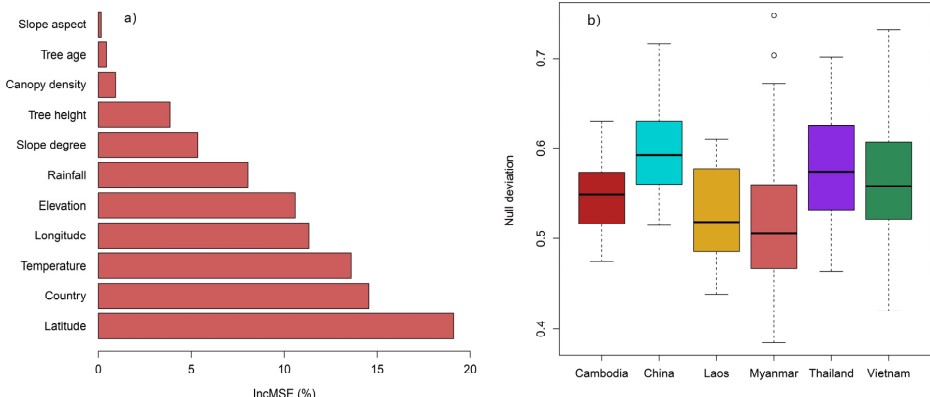


**Figure 6** Divers of plantation community or rubber plantation in GMS (a: Predictions of the
importance of environmental variables based on random forests; b: Boxplots showing the
relative changes in deterministic and stochastic processes assessed by null deviation analysis.
A null deviation close to zero suggests that stochastic processes are more important in
structuring the community, whereas a null deviation larger than zero indicates that
deterministic processes are more important)






232   In order to clarify whether exotic species can reduce plant diversity, we analyzed the

233 relationship between the dominance of exotic species and the species richness in the plot. In

234 view of the fact that *C. odorata* and *M. pudica* are the two most common exotic species in

235 rubber plantations (Figure 7a) the two species were selected for analysis. The importance

236 values of exotic species *C. odorata* (Figure S2a) and *M. pudica* (Figure S2k) were negatively

237 correlated with species richness, suggesting that exotic species with high dominance will

238 reduced rubber plantation diversity. Exotic species richness was positively correlated with

239 species richness (Figure 7c). Richness of communities where *C. odorata* (*M. pudica*) was

240 present was not lower than those where it was absent (Figure 7b). In sum, diversity of the

241 community was reduced only when the dominance of exotic species was high.

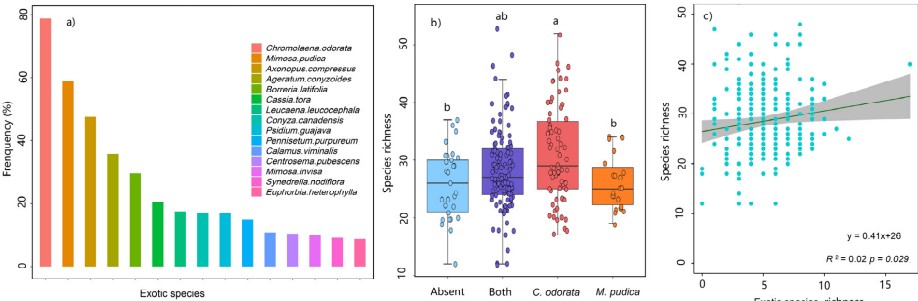

243 **Figure 7** Effects of exotic species on plant diversity of rubber plantations in the GMS (a:

244 Frequency of the most common exotic species; b: Richness comparison of different

245 communities (sky blue bar: plots without *C. odorata* and *M. pudica*; blue bar: plots with both

246 *C. odorata* and *M. pudica*; red bar: plots only with *C. odorata*; yellow bar: plots only with *M.*

247 *pudica*) c: relationship between exotic species richness of given plot and species richness of

248 given plot)



## 4. Discussion

### *4.1 Traditional ecological theory, in part, explain diversity patterns of rubber plantations*

Rubber plantations constitute one of the most important agro-ecosystems of tropical regions

and play an important role in their carbon budgets (Chen et al., 2020). In artificial forests such

as rubber plantations, there is no doubt that management measures and agricultural intensity

are two most important factors affecting plant diversity. For example, herbicide application

causes low diversity of understory plants. This is especially true of rubber plantations of

Vietnam (Figure S1f). Usually, species richness increases with lower latitude. However, we

found that species richness increases with higher latitude, peaking at about 25 degrees which

was the highest latitude we studied. We were surprised to find that understory plant diversity

of artificial rubber plantations increased with latitude and decreased with longitude, similar to

that of the global diversity patterns (Rohde 1992; Perrigo et al., 2013). This patterns is first

widely observed in regional rubber plantations. It has been suggested that the latitudinal

diversity gradient could be caused by habitat variables such elevation and slope degree. Here,

we uncovered a positive correlation between elevation (slope degree) and latitude (Figure S3).

We also found that there was a negative relationship between rainfall and longitude. This may

suggest that plant diversity increased with latitude mainly because elevation and slope increase

with latitude, and the diversity decrease with longitude was due to decreased rainfall in the

study area. Our results also showed that tree height positively correlated with understory

diversity. The possible explanation for this phenomenon is that higher height possible means

there are more space under the plantation and make more shade tolerant species survival.



Plant diversity of north Laos and south China was relatively higher than other countries. This
observation may be due to the large variation in elevation (for north Laos, elevation ranges
from 300 to 900 m; for south China, elevation ranges from 100-1100 m) in these areas, which
translates into greater environmental heterogeneity. We also found greater plant diversity at
higher elevations (Figure 5c), which may be caused by the reduced agricultural activities on
those terrains. It is not easy for farmers to clear understory plants on the steep slopes of rubber
plantations at high elevation; thus high slope degree indirectly results in low agricultural
intensity. In addition, greater slope may increase environmental heterogeneity and expand
niche space (Morrison-Whittle and Goddard, 2015). In sum, the traditional ecological
hypotheses, such as the latitudinal diversity gradient and niche partitioning, could also
contribute to explaining diversity patterns of artificial rubber plantations. However,
comparison of the diversity between rubber plantation and nearby natural forest needs further
research.
*4.2 Not all exotic plants cause the loss of plant diversity in rubber plantations*
Rubber plantation expansion and intensification has occurred in many regions that are key for
biodiversity conservation. Monoculture plantations have been promoted to restore the world's
forested areas, but have done little to slow the loss of biodiversity (Zhang et al., 2021). A
recent study shows exotic plants account for ~17% and ~35% of the total importance value
indices of natural and human-modified ecosystems, respectively (Chandrasekaran et al.,
2000). Here, in rubber plantations, exotic plants made up roughly 12% of the total recorded
species and 22.80% of the coverage. *C. odorata* is a noxious perennial weed in many parts of
the world (Kushwaha et al., 1981), and it is unsurprising that it was recorded in almost all





plantation plots in our study. *M. pudica*, the "sensitive plant", is a worldwide, pan-tropical
invasive species (Melkonian et al., 2014). *M. pudica*, as many tropical grasses and herbs, is
tolerant of low pH (Humphreys 1997, Paudel 2018), which explains its ubiquity in acidic
rubber plantation soil.
***4.3 Suggestions to improve rubber plantation plant diversity***
Previous study showed that rubber cultivation not only affect plant diversity (Hu et al., 2016),
but also affects the soil fauna, bird diversity as well as bat diversity. For example, compared
with natural forest, rubber plantations reduces the taxa richness of earthworm (Chaudhuri et
al., 2013), about 30% nematode taxa richness (Xiao et al., 2014), 50-60 % bird species
(Aratrakorn et al., 2006; Li et al., 2013) and bat species (Phommexay et al., 2011).
Many tropical regions, especially the GMS, contain hotspots of biodiversity that are
threatened by agriculture (Delzeit et al., 2017, Egli et al., 2018; Shackelford et al., 2014, Kehoe
et al., 2017). We must balance conservation with the economic goals of the GMS where the
livelihood of many people rely on rubber plantations. Well-managed forests can alleviate
poverty in rural areas, as outlined by the United Nations Sustainable Development Goals
(Lewis et al., 2019). Harvesting rubber latex may be the only way for many rural populations
to generate stable income, but rubber production does not assume a comfortable living. For
example, in Cambodia, many school children have to harvest rubber latex after classes to
support their families (Figure S1e). Due to the low Human Development Index, people in GMS
must prioritize supporting their families before protecting the biodiversity of the region. It
cannot fall squarely on local peoples to solve biodiversity crises.
In poor areas, we cannot just talk about ecological goals without first understanding local



cultures and economies. The rubber industry has not made preserving forest biodiversity a
major priority, and has struggled to meet conservation goals while minimizing economic loss
(Lan et al., 2017). Our results showed that diversity of different countries varies significantly
due to the variation in agricultural practice. In Vietnam, where diversity was low, rubber
farmers clear the understory to facilitate tapping and other production activities (Figure S1f).
The lower diversity in Cambodia may be due to the rubber plantations in northeastern which
are managed by Vietnamese rubber companies. Vietnam and Cambodia, and regions that allow
similar practices, augment the conflicts between agricultural production activities and
biodiversity conservation. More effort must be given to balance agricultural production with
biodiversity conservation goals in these regions. Thus, more innovative management measures,
such as cease of weeding and herbicide application (He and Martin, 2015), must be
implemented to improve the biodiversity of rubber plantations, so as to promote the
biodiversity of the region. Previous study conducted in India demonstrated that a no-weeding
practice in mature rubber plantations did not affect rubber yield (Abraham and Joseph, 2016).
A similar study conducted in China also showed that natural management strategies can
improve biodiversity without reducing latex production (Lan et al., 2017d). There is strong
evidence that adopting more natural management strategies improves plant diversity without
reducing latex production (Lan et al., 2017d). Thus, more innovative management measures
must be implemented to improve the plant diversity of rubber plantations, so as to promote the
biodiversity of the region.

**5.  Conclusion**



We provide a large regional study on the plant diversity of rubber plantations in a global
biodiversity hotspot. Plant diversity followed global trends with respect to longitude, latitude,
and altitude. Thus, artificial rubber plantation communities still conform to some common
ecological patterns. Exotic species were very common in rubber plantations, especially where
agricultural intensity was strong. However, not all exotic species directly drive the loss of
biodiversity. Only higher dominance of some exotic species were associated with a loss of
plant diversity within rubber plantations. We must make greater efforts to balance agricultural
production with conservation goals in this region, particularly in Vietnams and Cambodia, to
minimize the loss of biodiversity.

**Code availability**
Not applicable
**Authors' contributions**
**Guoyu Lan**: Conceptualization, Methodology, Writing, Reviewing and Editing; **Bangqian**
**Chen**: Methodology, Reviewing and Editing, **Chuan Yang, Rui Sun, Bangqian Chen,**
**Zhixiang Wu and Xicai Zhang**: Investigation
**Competing interests**
The authors declared that they have no conflicts of interest to this study.
**Disclaimer**
Publisher's note: Copernicus Publications remains neutral with regard to jurisdictional claims
in published maps and institutional affiliations.

**Acknowledgements**



We thank the help of the Rubber Research Bureau of Thailand, Perennial Crop Research
Institute of Myanmar, the Rubber Research Institute of Cambodia, Vietnam Agricultural
University, and the Yunnan Rubber Group. We would like to thank Prof. Fangliang He at
University of Alberta and Dr. Ian Gilman at Yale University for his assistance with English
language and grammatical editing.

**Funding**
This large-scale field investigation was supported by National Natural Science Foundation of
China (42071418), the Lancang-Mekong international cooperation project of the Ministry of
Foreign Affairs (081720203994192003) and the Earmarked Fund for China Agriculture
Research System (CARS-33-ZP3).

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





**Figure captions**
**Figure 1** Sampling plot localities within rubber plantations in GMS
**Figure 2** Distribution maps of two common exotic species (a: *Chromolaena odorata*, b:
*Mimosa pudica*) of rubber plantation in the GMS (circle size is proportional to importance
value)
**Figure 3** Plant species diversity of rubber plantations across countries in the GMS (a: species
richness; b: Shannon diversity; c: Exotic species richness).
**Figure 4** Beta diversity of rubber plantations in the GMS (a: PCoA ordination plot, b:
Whittaker's beta diversity (circle size is proportional to beta diversity value))
**Figure 5** Linear regressions of species richness of rubber plantation with environmental
variables (a: latitude; b: longitude; c: elevation; d: rainfall; e: temperature; f: slope degree; g:
tree age, h: tree height; i: canopy density)
**Figure 6** Divers of plantation community or rubber plantation in GMS (a: Predictions of the
importance of environmental variables based on random forests; b: Boxplots showing the
relative changes in deterministic and stochastic processes assessed by null deviation analysis.
A null deviation close to zero suggests that stochastic processes are more important in
structuring the community, whereas a null deviation larger than zero indicates that
deterministic processes are more important)
**Figure 7** Effects of exotic species on plant diversity of rubber plantations in the GMS (a:
Frequency of the most common exotic species; b: Richness comparison of different
communities (sky blue bar: plots without *C. odorata* and *M. pudica*; blue bar: plots with both
*C. odorata* and *M. pudica*; red bar: plots only with *C. odorata*; yellow bar: plots only with *M.*



*pudica*) c: relationship between exotic species richness of given plot and species richness of
given plot)




582       Table 1 Composition of plants of rubber plantations in GMS

| Types | No. of | Lifeform (%) | No. of families | No. of genera | No. of species |
|---|---|---|---|---|---|
| Ferns | 76 (8.00) | Non-woody plant | 86(38.05) | 278(45.65) | 445(46.89) |
| Gymnosperms | 3 (0.32) | Liana | 32(14.16) | 62(10.18) | 101(10.64) |
| Angiosperm | 870 (91.68) | Shrub | 42(18.58) | 118(19.38) | 192(20.23) |
| | | Tree | 66(29.20) | 151(24.79) | 211(22.23) |
| Total | 949 (100) | Total | 226(100.00) | 609 | 949 |
