# Peer review of "Main drivers of plant diversity patterns of rubber plantations in the"

_Biogeosciences, 2021_

## Author Comment (AC2)

Journal title: Biogeosciences

Title: Main drivers of plant diversity patterns of rubber plantations in the Greater Mekong Sub-region

On behalf of my co-authors, we appreciate the reviewer very much for his positive and constructive comments and suggestions on our manuscript entitled "Network complexity of rubber plantations is lower than tropical forests for soil bacteria but not fungi" submitted to SOIL. We have studied the reviewer' comments carefully and made the revision according the comments of the reviewers. The following are major changes (in blue) in the revised MS and responses to the comments.

1. There are lack of strong connection between the review in the introduction and the questions raised by the authors. The questions seems come out from nowhere. For example, there is no review about how exotic species influence plant diversity in rubber plantation, and no description on effects of deterministic and stochastic processes on artificial plantation.

Response: In the revised manuscript, we deleted the contents related to effects of deterministic and stochastic processes on artificial plantation. We have read some literatures on the impact of exotic species on plant diversity as well as the drivers for plant diversity. We have rewritten the introduction to make it logical.

**Reference:**

Stadler, J, Trefflich, A, Klotz, S, et al. (2000). Exotic plant species invade diversity hot spots: the alien flora of northwestern Kenya. Ecography. 23(2):169-176.

Stohlgren, T.J. , Binkley, D., Chong, G.W., et al. (1999). Exotic plant species invade hot spots of native plant diversity. Ecological Monographs, 69(1):25-46.

Han, Z.Q., Liu, T. , Wang, T. , Liu, H.F., Li, B. L. (2020). Quantification of water resource utilization efficiency as the main driver of plant diversity in the water-limited ecosystems. Ecological Modelling, 429, 108974

Xu, H.W., Liu, Q., Wang, S.Y., Yang, G.S., Xue, S. (2021). A global meta-analysis of the impacts of exotic plant species invasion on plant diversity and soil properties. Science of the Total Environment, 810, 152286

Nottingham, A., Fierer, N., Turner, B., Whitaker, J., Ostle, N., Mcnamara, N., et al. (2016). Temperature drives plant and soil microbial diversity patterns across an elevation gradient from the Andes to the Amazon. https://doi.org/10.1101/079996

Soliveres, S., Maestre, F.T. (2014). Plant-plant interactions, environmental gradients and plant diversity: a global synthesis of community-level studies. Perspectives in Plant Ecology Evolution & Systematics, 16(4), 154-163.

Soons, M.B., Hefting, M.M., Dorland,E., Lamers, L.P.M., Versteeg, C., Bobbink, R. (2017). Nitrogen effects on plant species richness in herbaceous communities are more widespread and stronger than those of phosphorus. Biological Conservation, 212, 390-397.

The authors review a lot of effects of rubber plantation on the plant diversity, however, this study is not about a comparison of diversity between rubber plantation and any other type of vegetation. Instead, all their data come from rubber plantation, their purpose is to find out the key factors in driving plant diversity within rubber plantations.

Response: Thanks for the comment. We have rewritten the introduction to make it logical.

2. The study methods are not clear.

a) There is contradiction between calculation of importance value and Shannon index. Out of importance value, the relative frequency is clearly computed based on the 240 plots, i.e. the whole study region (GMS), however, the Shannon index is used to quantify the diversity a focal plot. In another words, the importance value is computed at metacommunity level, but the Shannon index is computed at local community level.

Response: Thanks for the comments. Sorry for the errors.

For local community, there is no frequency data, therefore importance value is defined as: $IV_j = RH_j + RD_j$. For metacommunity, importance value is defined as: $IV_j = RF_j + RH_j + RD_j$.

b) The beta diversity index, Whittaker's β diversity, is not pairwise index. But Fig. 3

indicates that it quantifies the beta diversity between countries.

Response: Thanks for the comments. In the revised manuscript, we deleted contents related to Whittaker's β diversity.

c) Because there are correlations between environmental variables, doing linear regression one environmental variable by environmental variable is not appropriate.

Response: Thanks for the comments. In the revised manuscript, we used multiple linear regression to find the relationship between species richness and environmental variables (see Figure 1).

[Figure]

**Figure 1 Predicting species richness by using multiple linear regression. The red point was the observed richness, the green solid line was the estimated richness, and the grey solid line was the 95% confidence interval. y: Richness, $x1$: Latitude, $x2$: Elevation, $x3$: Slope, $x4$: Age, $x5$: Height, $x6$: Rainfall, $x7$: Temperature**

d) It not clear that Random forests is used to model alpha or beta diversity.

Response: Thanks for the comments. We have clarify it in the revised manuscript. Random forests is used to model alpha diversity.

e) I suggest use NMDS to present the ordination difference between the countries as NMDS maximum preserve the original dissimilarity between plots in two axes. The

PCoA here only explain about 13% of the community variation.

Response: Thanks for the comments. We used NMDS to present the ordination difference between the countries, however the stress is greater than 0.2 (Figure 2), therefore NMDS is not proper for this study.

Because plant diversity of rubber plantation is greatly affected by intensity of management, the explained percentage by PCoA as well as other methods is very small. Our RDA analysis results show that the explained percentage is 17.32%. Therefore, in the revised manuscript we still use PCoA to analyze beta diversity among countries. At the same time, the results of analysis of similarity showed that there were significant differences in plant composition among countries (Figure 3).

[Figure]

**Figure 2 Non-metric multidimensional scaling analysis (NMDS) based Bray-Curtis distance for plant communities of rubber plantation in the GMS.**

[Figure]

**Figure 3 Principal coordinate analysis (PCoA) based on Bray-Curtis distance and analysis of similarity for plant composition of rubber plantation in the GMS**

f) Personally, I don't think the null deviation can represent the effect between deterministic and stochastic processes due to many reasons which I don't want to refer to the details.

Response: Thanks. We have deleted null deviation and related contents.

3. Line 259-261 is against our common consensus that plant diversity decrease with latitude increasing.

Response: Latitudinal gradients in species-diversity are well known. They usually consist of a fairly regular increase in the numbers of species of some higher taxon from the poles to the equator. However, latitudinal gradients are known in which maximum diversity does not occur near the equator (Stehli, 1968). Our results was similar to that of the global diversity patterns that maximum diversity does not occur near equator.

4. Last but not least, I think the way and intensity of management between different plots or countries might contribute to plant diversity in rubber plantation greatly. However, the data might be not available. The effect way and intensity of management might intertwined with the measured factors in this study, how the author disentangle the effects between the factors is critical to the results and thus the conclusions.

Response: Thanks for the comments. Data about the intensity of management might be not available, thus it is difficult to distinguish the contribution of management to the diversity. In the revised manuscript, redundancy analysis were used to calculate the total contribution of main environmental factors to species composition, and most of the rest might be the contribution of intensity of management (see Figure 3).

[Figure]

[Figure]

**Figure3 Redundancy analysis of plant community composition of rubber plantation in the GMS (A: RDA ordination, B: proportion of explained)**

5. Please revised the introduction and methods and thus the results and discussion. Moreover, there is no analysis and result can support the so called suggestions in the discussion part, unless the author provide evidences in terms of their own analyses and results. Or else, the author should reduce the length of the suggestions.

Response: Thanks for the comments. We have revised the introduction, method, results as well as the discussion to make the paper logical. We fully agree with the reviewer's comments and have reduce the length of suggestions in the discussion part.

---

## Author Response (AR1)

Manuscript number: BG-2021-335

Journal title: Biogeosciences

Title: Main drivers of plant diversity patterns of rubber plantations in the Greater Mekong Sub-region

On behalf of my co-authors, we appreciate Dr. Li Yuwu very much for his positive and constructive comments and suggestions on our manuscript entitled "Main drivers of plant diversity patterns of rubber plantations in the Greater Mekong Sub-region" submitted to Biogeosciences. We have studied the reviewer' comments carefully and made the revision according the comments of the reviewers. The following are major changes (in blue) in the revised MS and responses to the comments.

General comments:

At present, there are a large number of reports focusing on the diversity of soil microorganisms in rubber plantations. Due to the plant diversity of rubber plantations largely varies with different management, it is difficult for us to find the driving factors for plant diversity of rubber plantations. This may be the main reason that there are few reports on the plant diversity of rubber plantations. I'm glad to see this report on the plant diversity of rubber plantations on such a large scale. This study revealed the driving factors for the plant diversity of rubber plantations. The exotic species in rubber plantations may influence the plant diversity, which we have paid little attention to before and the results can also attract the attention of readers. This article is also well organized and written. It is recommended to publish it after a minor revision.

Specific comments:

Line 29: "These gradients could be explained by the traditional ecological theories" this sentence is not clear. Traditional ecological theories? Niche hypothesis? Climate hypothesis?

Response: Thanks for the comments. We have rewritten the abstract.

Line 34: "In conclusion, not only environmental factors (such as elevation and

latitude)….".Here, I think the variations of elevation is more important for plant diversity because it is related to habitat heterogeneity.

Response: Thanks for the comments. We have rewritten the abstract.

Line 36- 38 "Much more effort should be made to balance agricultural production with conservation goals in this region, particularly to minimize the diversity loss in Vietnam and Cambodia" I found this sentence to jump out of the abstract without previous reference. Please make it logical.

Response: Thanks for the comments. We have rewritten the abstract.

Line 118. "We started the investigation only after the guide (local people) asked the farmer's consent". This sentence is unnecessary, please delete it.

Response: Thanks for the comments. According the comments of associate editor, we still retained it.

Line 132-134: "According to Sun's ( 2000) classification of plant uses, species were divided into medicinal plants, edible plants, economic plants, forage plants, ornamental plants, ecological plants and others (unknown use)". Plant uses have little relation with diversity. Please reconsider deleting it or not.

Response: Thanks for the comments. Deleted the related contents including Tables S3.

Line 303: "Many tropical regions…" delete "Many".

Response: Thanks for the comments. Deleted.

Line 251: "Traditional ecological theory…." Traditional ecological theories mean what? Please clarify.

Response: Thanks for the comments. Deleted.

Journal title: Biogeosciences

Title: Main drivers of plant diversity patterns of rubber plantations in the Greater Mekong Sub-region

On behalf of my co-authors, we appreciate the reviewer very much for his positive and constructive comments and suggestions on our manuscript entitled "Network complexity of rubber plantations is lower than tropical forests for soil bacteria but not fungi" submitted to SOIL. We have studied the reviewer' comments carefully and made the revision according the comments of the reviewers. The following are major changes (in blue) in the revised MS and responses to the comments.

1. There are lack of strong connection between the review in the introduction and the questions raised by the authors. The questions seems come out from nowhere. For example, there is no review about how exotic species influence plant diversity in rubber plantation, and no description on effects of deterministic and stochastic processes on artificial plantation.

Response: In the revised manuscript, we deleted the contents related to effects of deterministic and stochastic processes on artificial plantation. We have read some literatures on the impact of exotic species on plant diversity as well as the drivers for plant diversity. We have made substantial changes to the introduction.

**Reference:**

Han, Z.Q., Liu, T., Wang, T., Liu, H.F., Li, B. L. Quantification of water resource utilization efficiency as the main driver of plant diversity in the water-limited ecosystems. Ecol. Model., 429, 108974, 2020.

Mccoy, E.D., Connor, E. F. Latitudinal gradients in the species diversity of north American mammals. Evolution, 34, 193-203, 1980.

Nottingham, A., Fierer, N., Turner, B.L., Whitaker, J., Ostle, N.J., Mcnamara, N.P., Bardgett, R.D., Leff, J.W., Salinas, N., Silman, M. R., Kruuk, L. E. B., Meir, P. Temperature drives plant and soil microbial diversity patterns across an elevation gradient from the Andes to the Amazon. Ecology, 99(11), 2455-2466, 2018.

Soons, M.B., Hefting, M.M., Dorland,E., Lamers, L.P.M., Versteeg, C., Bobbink, R.

Nitrogen effects on plant species richness in herbaceous communities are more widespread and stronger than those of phosphorus. Biol. Conserv., 212, 390-397, 2017.

Stadler, J, Trefflich, A, Klotz, S, Brandl. R.. Exotic plant species invade diversity hot spots: the alien flora of northwestern Kenya. Ecography. 23(2):169-176. 2000.

Stohlgren, T.J. , Binkley, D., Chong, G.W., Kalkhan, M. A., Schell, L. D., Bull, K. A., Otsuki, Y., Newman, G., Bashkin, M., Son, Y. Exotic plant species invade hot spots of native plant diversity. Ecol. Monogr., 69(1): 25-46. 1999.

Xu, H.W., Liu, Q., Wang, S.Y., Yang, G.S., Xue, S. A global meta-analysis of the impacts of exotic plant species invasion on plant diversity and soil properties. Sci. Total Environ., 810, 152286, 2022.

The authors review a lot of effects of rubber plantation on the plant diversity, however, this study is not about a comparison of diversity between rubber plantation and any other type of vegetation. Instead, all their data come from rubber plantation, their purpose is to find out the key factors in driving plant diversity within rubber plantations.

Response: Thanks for the comment. We have rewritten the introduction to make it logical.

2. The study methods are not clear.

a) There is contradiction between calculation of importance value and Shannon index. Out of importance value, the relative frequency is clearly computed based on the 240 plots, i.e. the whole study region (GMS), however, the Shannon index is used to quantify the diversity a focal plot. In another words, the importance value is computed at metacommunity level, but the Shannon index is computed at local community level.

Response: Thanks for the comments. Sorry for the errors.

For local community, there is no frequency data, therefore importance value is defined as: $IV_j = RH_j + RD_j$. For metacommunity, importance value is defined as: $IV_j = RF_j + RH_j + RD_j$.

b) The beta diversity index, Whittaker's β diversity, is not pairwise index. But Fig. 3 indicates that it quantifies the beta diversity between countries.

Response: Thanks for the comments. In the revised manuscript, we deleted contents related to Whittaker's β diversity.

c) Because there are correlations between environmental variables, doing linear regression one environmental variable by environmental variable is not appropriate.

Response: Thanks for the comments. In the revised manuscript, we used multiple linear regression to find the relationship between species richness and environmental variables (see Figure 6a).

d) It not clear that Random forests is used to model alpha or beta diversity.

Response: Thanks for the comments. We have clarify it in the revised manuscript. Random forests is used to model alpha diversity.

e) I suggest use NMDS to present the ordination difference between the countries as NMDS maximum preserve the original dissimilarity between plots in two axes. The PCoA here only explain about 13% of the community variation.

Response: Thanks for the comments. We used NMDS to present the ordination difference between the countries, however the stress is greater than 0.2, therefore NMDS is not proper for this study.

Because plant diversity of rubber plantation is greatly affected by intensity of management, the explained percentage by PCoA as well as other methods is very small. Our RDA analysis results show that the explained percentage is 17.32%. Therefore, in the revised manuscript we still use PCoA to analyze beta diversity among countries. At the same time, the results of analysis of similarity showed that there were significant differences in plant composition among countries (Figure 3b).

f) Personally, I don't think the null deviation can represent the effect between deterministic and stochastic processes due to many reasons which I don't want to refer to the details.

Response: Thanks. We have deleted null deviation and related contents.

3. Line 259-261 is against our common consensus that plant diversity decrease with

latitude increasing.

Response: Latitudinal gradients in species-diversity are well known. They usually consist of a fairly regular increase in the numbers of species of some higher taxon from the poles to the equator. However, latitudinal gradients are known in which maximum diversity does not occur near the equator (Stehli, 1968). Our results was similar to that of the global diversity patterns that maximum diversity does not occur near equator.

Reference:

Stehli, G.G. Taxonomic diversity gradients in pole location: the Recent model, p, 168-227.1968.

4. Last but not least, I think the way and intensity of management between different plots or countries might contribute to plant diversity in rubber plantation greatly. However, the data might be not available. The effect way and intensity of management might intertwined with the measured factors in this study, how the author disentangle the effects between the factors is critical to the results and thus the conclusions.

Response: Thanks for the comments. Data about the intensity of management might be not available, thus it is difficult to distinguish the contribution of management to the diversity. In the revised manuscript, redundancy analysis and multiple regression were used to calculate the total contribution of environmental factors to species composition and diversity. And most of the rest might be the contribution of intensity of management. Our results showed that RDA analysis only explained 18.65% of the variation of community compositions, and multiple linear regression only explained 32.27% of the variation of plant diversity. Most of the unexplained variation are caused by management intensity and measures.

5. Please revised the introduction and methods and thus the results and discussion. Moreover, there is no analysis and result can support the so called suggestions in the discussion part, unless the author provide evidences in terms of their own analyses and results. Or else, the author should reduce the length of the suggestions.

Response: Thanks for the comments. We have made substantial changes to the introduction, method, results as well as the discussion to make the paper logical. We fully agree with the reviewer's comments and have reduce the length of suggestions in the discussion part.

---

## Author Response (AR2)

Manuscript number: BG-2021-335

Journal title: Biogeosciences

Title: Main drivers of plant diversity patterns of rubber plantations in the Greater Mekong Sub-region

On behalf of my co-authors, we appreciate Associate Editor Ben Bond-Lamberty very much for his positive and constructive comments and suggestions on our manuscript entitled "Main drivers of plant diversity patterns of rubber plantations in the Greater Mekong Sub-region" submitted to Biogeosciences. We have studied the comments carefully and made the revision according the comments of the Associate Editor. The following are major changes (in blue) in the revised MS and responses to the comments.

**Comments to the author**:

Both reviewers have re-read the manuscript and find that your revisions have significantly improved the MS, while addressing all their major concerns. I have read it as well and concur. There do remain many small issues of language and proper citations to fix -- please see list below -- but this should not take much work.

1.Line 26: "diversity, followed by China"

Response: Done.

2.L. 30: "Random Forest analyses" (note capitalized Random Forest, the name of the algorithm; please change throughout ms)

Response: Done.

3. L. 55: "Previous studies have shown that"

Response: Done.

4. L. 68: "Previous studies have also demonstrated"

Response: Done

5. L. 72, 107, 237, 244: start new paragraph - either because new idea, or too long

Response: Done

6. L. 82: "To test"

Response: Done

7. L. 104: "and thus"

Response: Done

8. L. 118: WorldClim2? Cite Fick and Hijmans 2017

Response: Thanks for the comments. Revised and cited the reference.

Reference:

Fick, S.E., Hijmans, R.J. (2017) WorldClim 2: new 1-m spatial resolution climate surfaces for global land areas. Int. J. Climatol., 37(12) https://DOI: 10.1002/joc.5086

9. L. 141: provide citation for Random Forest (usually Breiman 2001, but could be something else)

Response: Thanks for the comments. Revised and cited the reference.

Reference:

Breiman, L. (2001) Random Forests. Machine Learning, 45, 5-32.

9. L. 144: cite R correctly (see "citation ()"), and provide versions used (both of R and all packages). Also cite the vegan package correctly.

Response: Thanks for the comments. Revised as follows and cited the below references.

In order to understand how plant compositions are structured by environmental factors, a redundancy analysis (RDA) for the importance value of species was carried out using the Vegan packages (version 2.5-7) (Oksanen et al., 2020) in R (version 4.04) environment (R Core Team, 2021). Statistical significance was assessed using Monte Carlo tests with 999 permutations.

Reference.

R Core Team (2021). R: A language and environment for statistical computing. R Foundation for Statistical Computing, Vienna, Austria. URL https://www.R-project.org/.

Oksanen, J., Blanchet, F.G., Friendly, M., Kindt, R., Legendre, P., McGlinn, D., Minchin, P.R., O'Hara, R. B., Simpson, G.L., Solymos, P., Stevens, M.H.H., Szoecs, E., Wagner, H. (2020). vegan: Community Ecology. Package. R package version 2.5-7.

10. L. 232: "latitude ranks second"

Response: Done

11. L. 235: "are consistent with a previous study"

Response: Done

12. L. 240: "One study suggested that"

Response: Done

13. L. 243: "from our previous study in which elevation"

Response: Done

14. L. 247-249: unclear sentence; please reword

Response: Revised as follows:

Anyway, temperature could largely contribute to explaining the latitudinal diversity gradient patterns of rubber plantations.

15. L. 266: "was reduced...enough; not all"

Response: Done

16. L. 268: "follows"

Response: Done